# Design of Compact and Broadband Polarization Beam Splitters Based on Surface Plasmonic Resonance in Photonic Crystal Fibers

**DOI:** 10.3390/mi13101663

**Published:** 2022-10-03

**Authors:** Chao Mei, Yuan Wu, Jinhui Yuan, Shi Qiu, Xian Zhou

**Affiliations:** 1Research Center for Convergence Networks and Ubiquitous Services, University of Science and Technology Beijing (USTB), Beijing 100083, China; 2State Key Laboratory of Information Photonics and Optical Communications, Beijing University of Posts and Telecommunications, Beijing 100086, China

**Keywords:** photonic crystal fiber, polarization beam splitter, surface plasmonic resonance

## Abstract

In this work, a polarization beam splitter (PBS) based on surface plasmonic resonance is proposed and realized in a designed photonic crystal fiber (PCF). The PCF consists of two kinds of air holes with different diameters. Two solid silica cores near the center of the PCF are established by removing the cladding air holes. A gold film is plated at the external surface of the central air hole of the PCF to excite the surface plasmonic resonance. In order to minimize the length and improve the operation bandwidth of the PBS, the influences of the transversal structural parameters of the PCF are investigated in the context of both *X* and *Y* polarization beams. It was found that a 123.6-μm-long PBS with an operation bandwidth of 314 nm could be realized after the global optimization of five structural parameters. The proposed PBS may have potential applications in micro-/nano-optical systems for sensing and communications.

## 1. Introduction

Photonic crystal fiber (PCF) [1], which is made up of a central hole and surrounding periodical air holes, has been widely used in linear and nonlinear optics since it was first invented [2]. Compared to conventional step-index silica fiber, the arrangements of the air holes in PCFs enable distinct optical characteristics, such as high birefringence [3], controllable flattened dispersion [4], and endless single-mode transmission [5]. Due to the flexible engineering of the effective refractive index, $n_{\mathrm{eff}}$, PCF has been commonly employed to design polarization beam splitters (PBSes) [6,7], sensors [8,9,10], and couplers [11]. Among them, PBSes are some of the most important optical elements in both sensing and telecommunications. This element can be utilized to divide one orthogonal polarized beam into two polarized beams, which are generally called *X*-polarized (*X*-pol) and *Y*-polarized (*Y*-pol) beams [12]. Compared to PBSes made up of conventional silica fiber, PBSes realized with a PCF structure show shorter beam-splitting lengths and higher extinction ratios [13]; thus, they have attracted much attention in recent years.

PBSes based on dual-core PCF (DCPCF) are taken as examples. In 2015, L. Jiang et al. proposed a PBS by using DCPCF, whose air holes were organized in a rhombus lattice. The resulting beam-splitting length was 4036 μm and the bandwidth was 430 nm, covering the whole telecommunication band [14]. However, its length was more than 10 times larger than that of other PBSes because of the relatively large beam-splitting distance. In 2016, J. Zi et al. proposed a 249-μm-long PBS whose beam-splitting bandwidth was 12 or 17 nm, depending on its central pump wavelengths, which were located at 1.31 or 1.55 μm, respectively [15]. While J. Zi et al. significantly reduced the length of the PBS, the bandwidth was quite narrow. The performance of a designed PBS can be defined by the ratio of operating bandwidth and the PBS length. The resulting parameter is denoted as σ, and a larger value of σ indicates better beam-splitting performance. The values of σ for Refs. [14,15] were $1\times 10^{-4}$ and $6.8\times 10^{-5}$. In 2017, H. Wang designed a 5112-μm-long PBS in a PCF with two magnetic liquid cores. The resulting extinction ratio and bandwidth could reach −158 dB and 189 nm, respectively [16]. However, the calculated σ was only $3.7\times 10^{-5}$, implying that the designed PBS was still too long. In 2018, Q. Xu designed a DCPCF-based PBS with a beam-splitting length of only 290 μm. While the elliptical and circular air holes alternately arranged in the *Y*-direction were beneficial for the length reduction, the resulting bandwidth was only 19.2 nm around the wavelength of 1.55 μm [17]. The calculated σ was $6.6\times 10^{-5}$, which can be improved further. For the sake of larger values of σ in terms of short PBSes, an ultrashort PBS based on DCPCF filled with liquid was further proposed [18]. The resulting PBS length was only 78 μm, but was accompanied by a relatively large bandwidth of 44 nm. The corresponding σ was $5.6\times 10^{-4}$. The PBS length can be further reduced by using PCFs with square lattices instead of hexagonal lattices. J. Lou et al. proposed a PBS based on DCPCF with square lattices [19]. The obtained beam-splitting length was only 47.26 μm, which was much shorter than that in previous work. More importantly, the bandwidth was improved to 104 nm, which was larger than that reported in Ref. [18]. The calculated σ of $2.2\times 10^{-3}$ is, to the best of our knowledge, the largest value so far. However, square-lattice PCFs face more difficulties in fabrication than hexagonal-lattice PCFs do. While the PBSes designed in Refs. [18,19] were shorter than 100 μm, further reducing the PBS length became almost impossible due to the limit of diffraction and differences in the refractive index. It is thus necessary to pay more attention to increasing the operating bandwidth of PBSes.

In this paper, we propose a compact and broadband PBS based on DCPCF consisting of seven-layer air holes. As stated above, these air holes can be divided into two categories according to the size of their diameters. In order to improve the bandwidth of the PBS, we introduce surface plasmonic resonance (SPR) in the DCPCF, which has not been fully explored in PBS design. The SPR has been widely studied in optical sensors, as it can effectively induce a sudden local variation in the refractive index. With this feature, polarized beam splitting based on SPR is possible because polarized beams are also sensitive to changes in the refractive index. Therefore, increasing the operating bandwidth of a PBS can actually be achieved by tuning the wavelength at which the SPR effect works. In PCF, the characteristic wavelength of SPR is closely related to the transversal structure of a PCF whose size is comparable to the wavelength, i.e., on the order of μm. In such a case, the $n_{\mathrm{eff}}$ of the PCF is very sensitive to the SPR effect. In other words, the characteristic wavelength of SPR can be effectively tuned by changing the transversal structural dimensions of PCFs. The structural influences on the PBS are discussed in detail below.

## 2. Design of the DCPCF

Considering the practicality of fabrication, we still adopted a traditional hexagonal-lattice PCF to design the PBS. Figure 1 shows the cross-section of the proposed DCPCF. The outermost blue ring is the perfect matched layer (PML). It can be used to reflect all rays back into the PCF and prevent energy loss in numerical simulations. The background material marked in gray is silica. Seven-layer air holes are arranged periodically. The spacing between two large air holes is denoted by Λ, while the spacing between a large and a small air hole is defined by $d_{2}$. The diameter of a large air hole is $d_{1}$. It should be noted that the air hole located at the center of the DCPCF is coated with a gold film, which is marked in yellow. Gold and silver films are the two most often used materials to excite the SPR effect due to their faster electron mobility. The diameter of the central air hole and the thickness of the gold film are denoted by *d* and *t*, respectively. The two air holes on the left- and right-hand sides of the central air hole are removed. As a result, two solid silica cores named A and B are formed to allow the propagation of two orthogonal polarized beams. Next, we will adjust these five parameters to study the beam-splitting length for different polarized beams.

In order to study the SPR effect $n_{\mathrm{eff}}$ of PCF, it is necessary to introduce the Sellmeier equation of silica, which is given as [20,21]:(1)$$n^{2}\left(\lambda \right)=1+\frac{A_{1}\lambda^{2}}{\lambda^{2}-B_{1}^{2}}+\frac{A_{2}\lambda^{2}}{\lambda^{2}-B_{2}^{2}}+\frac{A_{3}\lambda^{2}}{\lambda^{2}-B_{3}^{2}},$$
where λ is the incident wavelength with units of μm, and $A_{1},A_{2},A_{3}$ and $B_{1},B_{2},B_{3}$ are constants, as shown in Table 1. The dispersion effect of air is neglected in this work, as it makes almost no contributions to the $n_{\mathrm{eff}}$ in the wavelength range considered.

Unlike in semiconductor materials, the dispersion effect of metal involves the modification of the dielectric constant. In most cases, the dielectric constant of a gold film can be defined by the Drude–Lorentz model as [22,23]
(2)$$\epsilon =\epsilon_{0}-\frac{\omega_{D}^{2}}{\omega (\omega +j\gamma_{D})}-\frac{\Delta \epsilon \Omega_{L}^{2}}{\left(\omega^{2}-\Omega_{L}^{2}\right)+j\Gamma_{L}\omega },$$
where $\epsilon_{0}$ = 5.9673 is the dielectric constant of gold, $\omega_{D}$ is the plasma frequency, $\omega_{D}/2\pi$ = 2113.6 THz, $\gamma_{D}$ is the damping frequency, $\gamma_{D}/2\pi$ = 15.92 THz, $\Omega_{L}$ is the central frequency of the Lorentz oscillator, $\Omega_{L}/2\pi$ = 650.07 THz, $\Gamma_{L}$ is the spectral width of the Lorentz oscillator, $\Gamma_{L}/2\pi$ = 104.86 THz, and ω is the angular frequency of the guided wave.

The coupling length *CL* is a vital parameter for describing the characteristics of a PBS, as the length of the PBS is determined by it. Generally, the CL is defined as the propagated distance when the power of *X* (or *Y*)-polarized light completely transfers from one core, i.e., A, to another core, i.e., B. The *CL* of *X* (or *Y*)-polarized light is called the *CLX* (or *CLY*), which is mathematically expressed as [24,25]
(3)$$CLi=\frac{\lambda }{2\left|n_{\mathrm{even}}^{i}-n_{\mathrm{odd}}^{i}\right|},\;\;\;i=X,Y,$$
where *i* represents the polarization states including both the *X*-pol and *Y*-pol directions, CLi is the coupling length of the *i*-pol direction, and $n_{\mathrm{even}}^{i}$ and $n_{\mathrm{odd}}^{i}$ are the $n_{\mathrm{eff}}$ of the even and odd modes, respectively. The ratio between the *CLY* and *CLX* can be obtained from [26]
(4)$$R=\frac{CLY}{CLX}.$$

When the beam propagates in a dual-core PCF, its energy will periodically transfer from core A to core B. We define that when *R* = 2 or 1/2, the incident beam is completely separated into two polarizations. Assuming that the beam is initially injected into core A, the output power of the beam in the *X*-pol or *Y*-pol directions in core A can be calculated by [27,28]
(5)$$P_{\mathrm{out}}^{i}=P_{\mathrm{in}}\cos^{2}\left(\frac{ßL_{p}}{2\mathrm{CLi}}\right),\;\;\;i=X,Y,$$
where $P_{\mathrm{in}}$ represents the input power injected into core A. When $P_{\mathrm{in}}$ is assumed to be 1, the output power $P_{\mathrm{out}}$ is a parameter that is normalized for easy understanding. $L_{p}$ is the propagation distance. The extinction ratio (ER) is used to evaluate the beam-splitting performance of the PBS. The ER of the output powers between the *X*-pol and *Y*-pol beams in core A can be defined as [29,30]
(6)$$ER=10\log_{10}\left(\frac{P_{\mathrm{out}}^{X}}{P_{\mathrm{out}}^{Y}}\right).$$

## 3. Influences of Structural Parameters on PBSes

To test whether the SPR effect can be stimulated in the designed DCPCF, we arbitrarily set the structural parameters as *d* = 1.1, $d_{1}$ = 1.3, $d_{2}$ = 1.0, and Λ = 2.0 μm, while the thickness of the gold film was assumed to be *t* = 50 nm. A numerical software, COMSOL, was employed to simulate the SPR in the designed DCPCF. Figure 2 displays the mode fields of the *X*-pol and *Y*-pol directions, respectively. Specifically, Figure 2a–c show the *X*-pol even mode, *X*-pol odd mode, and *X*-pol second SPR mode, respectively, when the pump wavelength was 1.239 μm. Similarly, Figure 2d–f show the *Y*-pol even mode, odd mode, and second SPR mode, respectively, when the pump central wavelength was 1.186 μm. As can be seen, the intensity of the even mode was much stronger than that of the odd mode and second SPR mode. More importantly, the energies of both the *X*-pol and *Y*-pol even modes were tightly confined in cores A and B, indicating that the beam splitting mostly occurred in the even mode. Figure 2 confirms that the SPR mode can be effectively excited in the designed DCPCF, but with low intensity. Nevertheless, this weak SPR effect can still modify $n_{\mathrm{eff}}$.

In order to show the influence of SPR on $n_{\mathrm{eff}}$ for both the *X*-pol and *Y*-pol beams, the variations in $n_{\mathrm{eff}}$ with wavelength for the two polarized beams are shown in Figure 3a,b. It should be noted that $n_{\mathrm{eff}}$ values of the X-pol odd mode and second SPR mode were exchanged when the wavelength was 1.239 μm, as shown in Figure 3a. Similarly, the $n_{\mathrm{eff}}$ values of the *Y*-pol odd mode and second SPR mode were exchanged when the wavelength was 1.186 μm, as shown in Figure 3b. This kind of sudden exchange resulted from the strong couplings between the odd mode and the second SPR mode. This coupling is unfavorable for polarization splitting, as the spatial profiles will be different after beam coupling. However, considering the low intensity of the odd and second SPR modes, the performance of the PBS will not be affected. The *CLX*, *CLY*, and *R* were also calculated for different wavelengths by using Equations (Equation 3) and (Equation 4), as shown in Figure 3c. The inflection point for every polarized beam corresponded to the mutations in Figure 3a,b. While the *X*-pol and *Y*-pol beams had inflection points at different wavelengths (1.18 μm and 1.25 μm, respectively), the resulting *R* approached 2 only after 1.3 μm. More importantly, the CL for the *Y*-pol beam was larger than 125 μm, which can be further optimized. In the following, the influences of the structural parameters of the DCPCF are optimized for R→2 under smaller CLs.

First, we considered the influence of the diameter of the central air hole, *d*. Figure 4a,b display the variations in *CLX* and *CLY* when *d* was increased from 0.9 to 1.3 μm. It can be seen that for all *d*, the *CLX* decreased with the wavelength before the inflection point and then remained almost unchanged after the inflection point. More importantly, the wavelength position of the inflection point was red-shifted when *d* was increased, indicating the reduction of the operation bandwidth. However, the reduced bandwidth was compensated by smaller *CLX*, which is beneficial when designing short PBSes. The *CLY* in Figure 4b showed a similar trend to that of *CLX*, with the small difference that the *CLY* was almost unchanged with both the wavelength and *d* between 1.6 and 1.7 μm. The corresponding *R* values were calculated according to Equation (Equation 4) and are shown in Figure 4c. Obviously, for the sake of larger values of *R*, increasing *d* was a good choice. The physical mechanism behind it is that changing the size of the central air hole essentially reshapes the distribution of the mode field. As a result, the mode couplings between the *X*-pol (*Y*-pol) odd mode and the *X*-pol (*Y*-pol) second SPR mode also vary. It should be noted that when *d* exceeds 1.2 μm, R>2, and the operation bandwidth is decreased. Therefore, *d* should be chosen carefully to balance the value and operation bandwidth of *R*.

We then investigated the influence of $d_{1}$ by keeping the other four parameters unchanged. Figure 5 depicts the variations in the *CLX*, *CLY*, and *R* with the wavelength when $d_{1}$ was increased from 1.1 to 1.5 μm. It can be seen from Figure 5a that the *CLX* remained unchanged with $d_{1}$ before the inflection points. After the inflection point, the *CLX* showed a tiny variation with the wavelength for a certain $d_{1}$. The minimum *CLX* under $d_{1}=1.5$
μm was 60 μm, which was similar to that in Figure 4a. In addition, the minimum *CLY* in Figure 5b was almost the same as that in Figure 4b. Consequently, the maximum *R* in Figure 5c was equal to 1.88, which was smaller than that in Figure 4c. However, the smaller value of *R* covered a much broader wavelength range, i.e., 1.24–1.7 μm. Therefore, $d_{1}$ has a small influence on the bandwidth and *R* compared with the influence of *d*.

Then, we studied the influence of $d_{2}$ on the *CL* and *R*. The relationship between the *CLX* and wavelength under different values of $d_{2}$ is shown in Figure 6a. It can be clearly seen that while similar variations in the *CLX* with wavelength could be observed, $d_{2}$ itself had almost no influence on the *CLX*. In contrast, the *CLY* in Figure 6b showed more changes with $d_{2}$. However, compared with that in Figure 5b, the variation in *CLY* was a little bit smaller. As a result, the influence of $d_{2}$ on the CL was smaller than that of $d_{1}$. However, it should be noted that the resulting *R* in Figure 6c showed a remarkable increment compared with that in Figure 5c when $d_{2}$ was increased. Actually, the maximum of R=1.95 and the operation wavelength covered 1.25–1.7 μm, which could be attributed to the completely invariant *CLX* when $d_{2}$ was increased. Compared with that in Figure 6c, the resulting *R* was quite close to the ideal value of 2.

While the optimization of $d_{2}$ was successful in obtaining a larger and broadband *R*, a short PBS is still required for a large σ. We further studied the influence of Λ on the CL and *R*. The simulation results are shown in Figure 7. Figure 7a,b depict the variations in *CLX* and *CLY* with the wavelength under different values of Λ. While both the *CLX* and *CLY* show similar trends of variation with the wavelength, the resulting minimum values of the *CLX* and *CLY* were only 40 and 65 μm, which were smaller than those in Figure 6a and Figure 6b, respectively. In general, the CL rapidly increased with Λ. This was because larger values of Λ made the mode coupling more difficult. It is worth noting that the value of *R* in Figure 7c showed an opposite trend compared with that in Figure 6c, i.e., the value of *R* decreased with Λ. Furthermore, *R* was more influenced by Λ, but at the expense of a reduced operation bandwidth. Nevertheless, changing $d_{2}$ and Λ could result in the same bandwidth and value of *R*.

Finally, the influence of *t* on the CL and *R* were investigated. Figure 8 shows the variations in the *CLX*, *CLY*, and *R* with the wavelength under different values of *t*. In Figure 8a,b, the *CLX* and *CLY* remained the same over the whole wavelength range under different values of *t*. This means that *t* had almost no influence on the CL. As a result, *t* also had a negligible influence on *R*, especially in the wavelength range of 1.3–1.7 μm. This is because small changes (5–20 nm) in *t* cannot influence the distribution of the optical mode. In Figure 8c, the largest values of *R* under different values of *t* showed tiny fluctuations around 1.9, which was smaller than the desired value of 2. Therefore, it is impossible to achieve R=2 by only adjusting *t*.

From above simulation results, we can see that only *d*, $d_{2}$, and Λ showed a significant relationship with the CL and *R*. Actually, *d* not only affected *R*, but also determined the wavelength position of the inflection point, namely, the operation bandwidth. In addition, Λ affected not only the value of *R*, but also the CL. In contrast, $d_{2}$ mainly affected the value of *R* and showed a slight influence on the resonance wavelength. In addition, the flatness of the *R* curve after the inflection point was also crucial because it determined the bandwidth of the PBS. In the five structural parameters, only $d_{1}$ enabled a flat *R* after the inflection point. Therefore, the optimization procedure can be summarized in the following three steps: first, adjusting *d* and $d_{2}$ to cause *R* to approach the ideal value of 2; second, changing the value of Λ for small CLs; third, controlling $d_{1}$ to improve the flatness of *R* after the inflection point. Following the above procedure, we obtained a set of optimized structural parameters of the DCPCF: *d* = 1.2 μm, $d_{1}$ = 1.3 μm, $d_{2}$ = 1.0 μm, Λ = 2.0 μm, and *t* = 50 nm.

## 4. Performance Analysis of the Optimized PBS

The *CLX*, *CLY*, and *R* calculated for the optimized PBS are shown in Figure 9 in the wavelength range of 1.0–1.8 μm. It can be seen from Figure 9a that both the *CLX* and *CLY* were well below 130 μm after the inflection point. More importantly, *R* slightly fluctuated around the ideal value of 2 after 1.35 μm. Figure 9b depicts the details of *R*. It is shown that *R* varied between 1.98 and 2.02 in the wavelength range of 1.39 and 1.6 μm. The maximum and minimum values of *R* were 2.015 and 1.985, which were very close to the ideal value of 2. When the pump wavelength was 1.405 or 1.568 μm, R=2. For the latter wavelength, the corresponding *CLX* and *CLY* were 123.6 and 61.8 μm, respectively. Therefore, 123.6 μm could be selected as the length of the proposed PBS.

By using Equations (Equation 5) and (Equation 6), Figure 10a could be obtained, presenting the relationship between the normalized output power of the *X*-pol and *Y*-pol beams along the propagation when the initial beam was injected into core A. It is shown that when the propagation distance was 123.6 μm, the normalized output power in the *X*-pol beam was 1, while that in the *Y*-pol beam was 0. At this moment, the *X*-pol and *Y*-pol beams were successfully separated into core A and core B, respectively. Figure 10b shows the relationship between the ER and the wavelength. Generally speaking, an ER larger than 20 dB can be considered an indication of successful splitting [31]. The bandwidth of a PBS is defined as the wavelength range in which the ER is larger than 20 dB. It can be seen from Figure 10b that the operation bandwidth was 314 nm and maximum ER was 78 dB at 1.57 μm. We then calculated the value of σ of our work and compared it with the values reported in other work. It can be seen from Table 2 that the value of σ in this work was the largest among all works, which strongly suggests that our PBS is better than others in terms of length and operation bandwidth.

## 5. Conclusions

In summary, we designed a compact and broadband PBS based on the SPR effect by using a DCPCF that we designed. By studying the influences of the transversal structural parameters, we found that, except for the thickness of the gold film, other parameters played important roles in the PBS’s performance. Based on these findings, we summarized an efficient optimization procedure for quickly designing short and broadband PBSes. A set of optimized structural parameters was obtained by using this optimization procedure. The resulting PBS showed a length of 123.6 μm, an operation bandwidth of 314 nm, and a maximum ER of 78 dB. Compared with other reported work, our PBS reached the best performance concerning the length and operation bandwidth. As the operation bandwidth is located in the telecommunication band, we believe that the proposed compact and broadband PBS has potential applications in miniaturized optical systems for sensing or communication.

## Figures and Tables

**Figure 1 micromachines-13-01663-f001:**
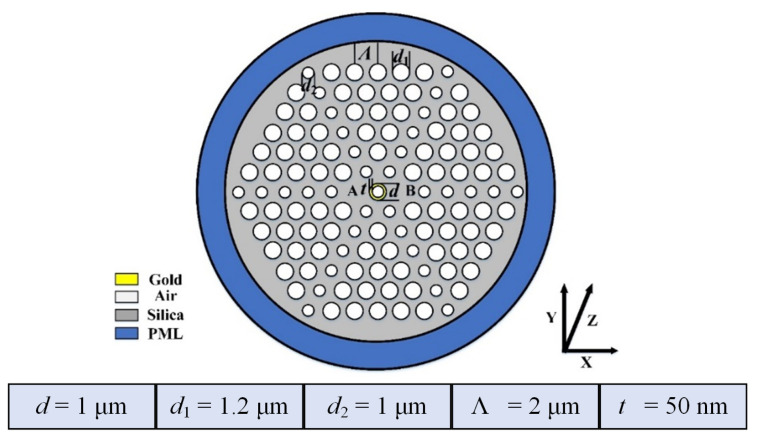
A sectional view of the proposed DCPCF with its structual parameters, $d,d_{1},d_{2},\Lambda$, and *t*.

**Figure 2 micromachines-13-01663-f002:**
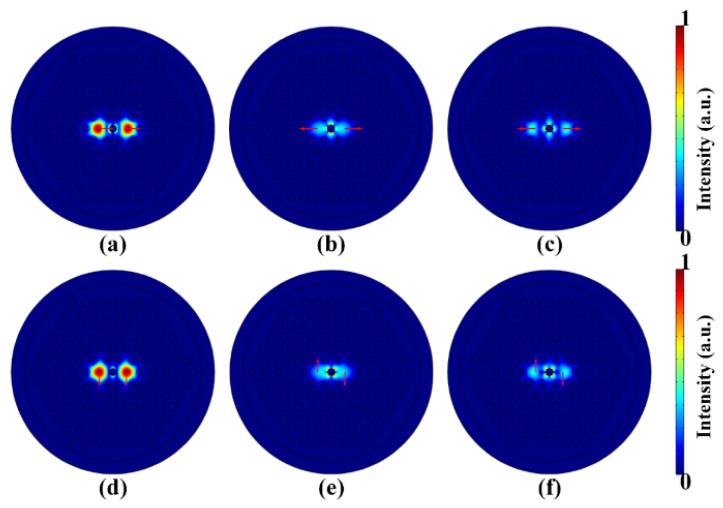
The mode fields of (**a**) the *X*-pol even mode, (**b**) *X*-pol odd mode, and (**c**) *X*-pol second SPR mode at the wavelength of 1.239 μm. The mode-field distribution of (**d**) the Y-pol even mode, (**e**) *Y*-pol odd mode, and (**f**) *Y*-pol second SPR mode at the wavelength of 1.186 μm.

**Figure 3 micromachines-13-01663-f003:**
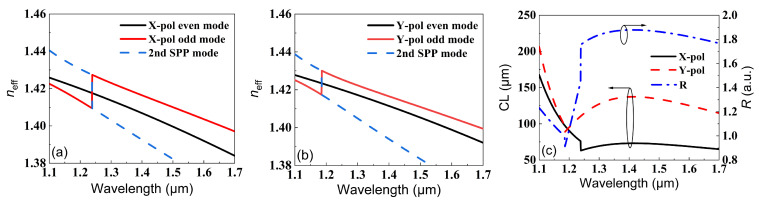
The effective refractive indexes of (**a**) the X-pol and (**b**) Y-pol modes. (**c**) The variations in *CLX*, *CLY*, and *R* with wavelength.

**Figure 4 micromachines-13-01663-f004:**
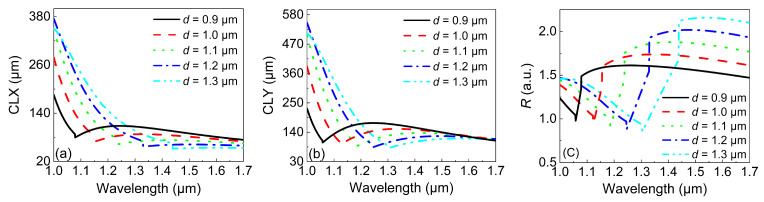
The variations in (**a**) the *CLX* and (**b**) *CLY* with the wavelength under different values of *d*. (**c**) The variations in *R* with the wavelength under different values of *d*.

**Figure 5 micromachines-13-01663-f005:**
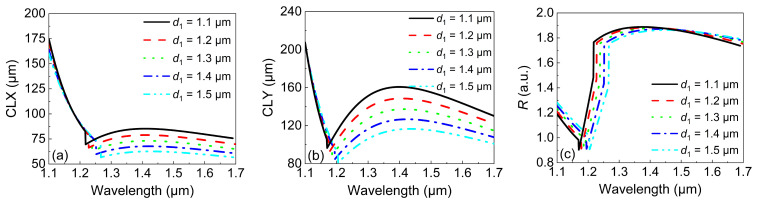
The variations in (**a**) the *CLX* and (**b**) *CLY* with the wavelength under different values of $d_{1}$. (**c**) The variations in *R* with the wavelength under different values of $d_{1}$.

**Figure 6 micromachines-13-01663-f006:**
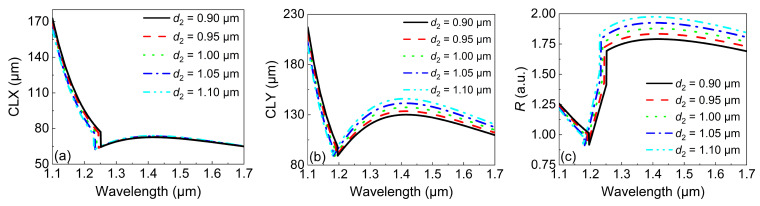
The variations in (**a**) the *CLX* and (**b**) *CLY* with the wavelength under different values of $d_{2}$. (**c**) The variations in *R* with the wavelength under different values of $d_{2}$.

**Figure 7 micromachines-13-01663-f007:**
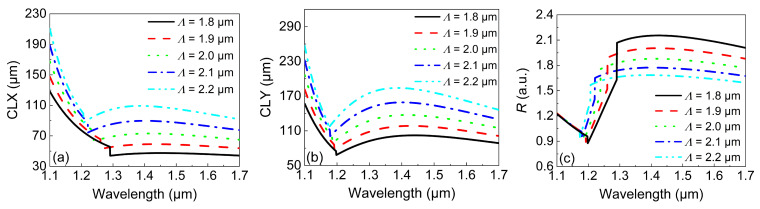
The variations in (**a**) the *CLX* and (**b**) *CLY* with the wavelength under different values of Λ. (**c**) The variations in *R* with the wavelength under different values of Λ.

**Figure 8 micromachines-13-01663-f008:**
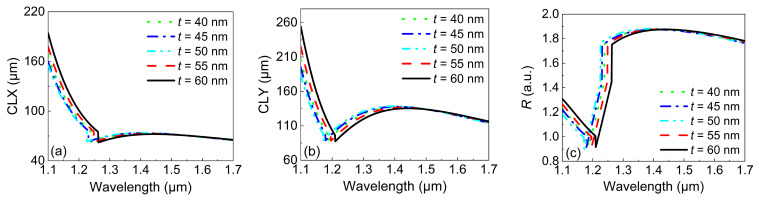
The variations in (**a**) the *CLX* and (**b**) *CLY* with the wavelength under different values of *t*. (**c**) The variations in *R* with the wavelength under different values of *t*.

**Figure 9 micromachines-13-01663-f009:**
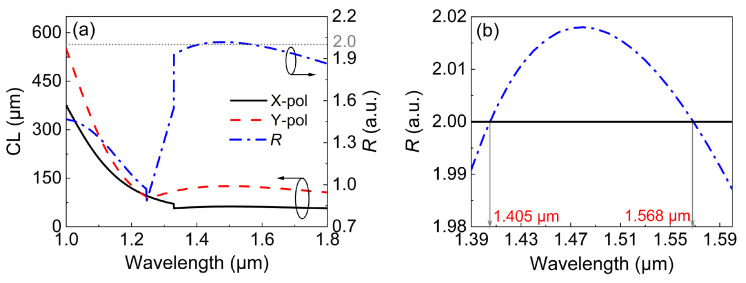
(**a**) The variations in *CLX*, *CLY*, and *R* with the wavelength for the optimized DCPCF. (**b**) Zooming in on *R* in the wavelength range of 1.39–1.6 μm.

**Figure 10 micromachines-13-01663-f010:**
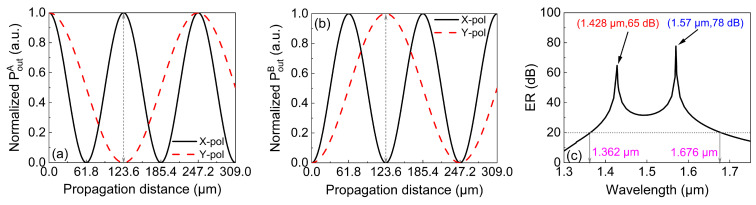
The relationship between the normalized output power and propagation distance in (**a**) core A and (**b**) in core B, and (**c**) the ER in core A between 1.3 and 1.75 μm with the optimized structural parameters.

**Table 1 micromachines-13-01663-t001:** Parameters for the Sellmeier equation of silica.

	Subscript 1	Subscript 2	Subscript 3
A	0.6961663	0.4079426	0.8974794
B	0.0684043 μm	0.1162414 μm	9.896161 μm

**Table 2 micromachines-13-01663-t002:** Comparison of the PBS performance between this work and other reported work.

Refs.	PBS Structure	Length (μm)	Bandwidth (nm)	Ratio σ
[14]	square PCF with gold wire	4036	430	$1\times 10^{-4}$
[15]	hexagonal PCF without gold	249	17	$6.8\times 10^{-5}$
[16]	magnetic-fluid-core PCF with gold film	5112	189	$3.7\times 10^{-5}$
[17]	PCF with elliptical holes and gold film	290	19.2	$6.6\times 10^{-5}$
[18]	PCF filled with liquid without gold	78	44	$5.6\times 10^{-4}$
[19]	octagonal PCF with gold film	47.26	104	$2.2\times 10^{-3}$
This work	hexagonal PCF with gold film	123.6	314	$2.5\times 10^{-3}$

## Data Availability

Data are available upon request from the corresponding author.

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
