# Peer review of "Design of Compact and Broadband Polarization Beam Splitters Based on Surface Plasmonic Resonance in Photonic Crystal Fibers"

_micromachines, 2022, doi:10.3390/mi13101663_

Round 1

Reviewer 1 Report

In the manuscript entitled "Design of compact and broadband polarization beam splitter based on surface plasmonic resonance in photonic crystal fibers", the authors designed a polarization beam plitter and optimized the geometric factor to optimize the operation in the given bandwidth. The work is nice done. I recommend the manuscript be published after make some minor edits:

(1) In figure 1, it's better to give a specific dimension example (i.e. d = XXX mm) for all the geometric parameters.

(2) In figure 1, what does PML stand for? Please define.

Reviewer 2 Report

The paper deals with a exhaustive numerical simulation for a compact polarization beam splitters.  

Therefore, they present a physical realization.

The paper is interesting and is suitable for publication.

Reviewer 3 Report

1.      Original Submission

1.1.Recommendation

Reject

2.      Comments to Author:

In this study, Mei et al. designed a polarization beam splitter (PBS) based on surface plasmonic resonance (SPR) in photonic crystal fiber (PCF). The manuscript on this is complete, however, as the authors mentioned that various researches about PBSs based on dual-core PCF (DCPCF) have been achieved and some of them includes the application of SPR. Particularly, the design of the PBS suggested in this work is almost identical to the design by Qu et al., (Sensors 2021, 21(2), 96; https://doi.org/10.3390/ s21020496), and the overall work flow is also very similar. Therefore, I couldn’t find any distinction and novelty from the work. Besides, some errors could be found in figures please double-check.
